ecology

$\alpha$ diversity, $\beta$ diversity, global change, homogenization, phylogenetic diversity

**Author for correspondence:**
Daijiang Li
e-mail: daijianglee@gmail.com

# Changes in taxonomic and phylogenetic diversity in the Anthropocene

Daijiang Li[1], Julian D. Olden[2], Julie L. Lockwood[3], Sydne Record[4], Michael L. McKinney[5] and Benjamin Baiser[1]

[1]Department of Wildlife Ecology and Conservation, University of Florida, Gainesville, FL 32611, USA
[2]School of Aquatic and Fishery Sciences, University of Washington, Seattle, WA 98105, USA
[3]Department of Ecology, Evolution and Natural Resources, Rutgers University, New Brunswick, NJ 08901, USA
[4]Department of Biology, Bryn Mawr College, Bryn Mawr, PA 19010, USA
[5]Department of Earth and Planetary Sciences, The University of Tennessee, Knoxville, TN 37996, USA

DL, 0000-0002-0925-3421; JDO, 0000-0003-2143-1187

To better understand how ecosystems are changing, a multifaceted approach to measuring biodiversity that considers species richness (SR) and evolutionary history across spatial scales is needed. Here, we compiled 162 datasets for fish, bird and plant assemblages across the globe and measured how taxonomic and phylogenetic diversity changed at different spatial scales (within site $\alpha$ diversity and between sites spatial $\beta$ diversity). Biodiversity change is measured from these datasets in three ways: across land use gradients, from species lists, and through sampling of the same locations across two time periods. We found that local SR and phylogenetic $\alpha$ diversity (Faith's PD (phylogenetic diversity)) increased for all taxonomic groups. However, when measured with a metric that is independent of SR (phylogenetic species variation, PSV), phylogenetic $\alpha$ diversity declined for all taxonomic groups. Land use datasets showed declines in SR, Faith's PD and PSV. For all taxonomic groups and data types, spatial taxonomic and phylogenetic $\beta$ diversity decreased when measured with Sorensen dissimilarity and phylogenetic Sorensen dissimilarity, respectively, providing strong evidence of global biotic homogenization. The decoupling of $\alpha$ and $\beta$ diversity, as well as taxonomic and phylogenetic diversity, highlights the need for a broader perspective on contemporary biodiversity changes. Conservation and environmental policy decisions thus need to consider biodiversity beyond local SR to protect biodiversity and ecosystem services.

## 1. Introduction

Habitat loss, biological invasions and climate change are progressively altering ecological systems across the world. At the global scale, we are arguably witnessing the greatest loss of biodiversity in the history of the Earth as extinctions have outpaced speciation events over the last century [1,2]. At local and regional scales, however, *in situ* taxonomic diversity within assemblages ($\alpha$ diversity) has increased or remained unchanged as the establishment of non-native species has either equalled or exceeded species losses [3–5], although how these changes have manifested at local spatial scales is debatable [6,7]. At the same time, studies that compare taxonomic diversity between human disturbed (i.e. land use change) and reference sites generally report declining local taxonomic diversity [8]. These divergent results suggest that multiple drivers must be simultaneously investigated to achieve a robust understanding of contemporary biodiversity change [7,9].

Species losses and gains have important implications for compositional similarity among assemblages (spatial $\beta$ diversity). The loss of endemic or rare species coupled with the establishment of cosmopolitan non-native species has resulted in taxonomic homogenization [10–13]. A recent meta-analysis of diversity time series provided evidence that despite no systematic loss of local species richness (SR), there has been extensive temporal changes in species composition (i.e. increased temporal $\beta$ diversity within sites) [5]. However, it

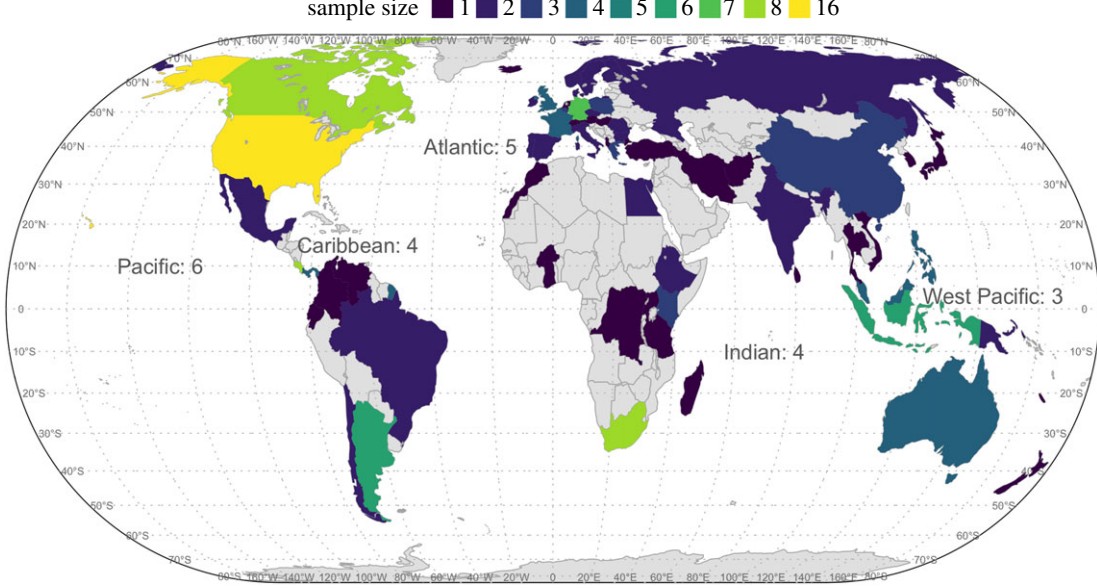

**Figure 1.** Geographical distribution of the 162 datasets used in this analysis. Colour shading reflects the number of datasets in each country, and the number of datasets per ocean is reported in the text. We did not plot exact locations because some datasets (e.g. species lists) cover a region and thus lack specific latitude/longitude locations. See Materials and methods for details and sources of the datasets. (Online version in colour.)

remains unknown if there have also been widespread changes in spatial $\beta$ diversity without loss of $\alpha$ diversity.

Contemporary changes in biodiversity, especially at global extents, have predominantly been investigated from a taxonomic perspective (i.e. species identity) [4,5,11]. Increasingly, the phylogenetic dimension of diversity is recognized as critical to understand the structure and functioning of species assemblages [14–16]. Assemblages with higher phylogenetic $\alpha$ and $\beta$ diversity are thought to be more resilient to disturbance and more productive because they possess evolutionary potential to adapt to changing environmental conditions [17–19]. Phylogenetic diversity can also serve as a proxy for the diversity of functional forms of organisms in a community [20], with higher functional diversity providing greater community stability [21]. Furthermore, rare taxa, which are often the focus of conservation efforts, tend to be evolutionarily distinct and, as a consequence, are often prioritized based on phylogenetic diversity metrics [22]. Taken together, mounting evidence suggests that understanding how forces of global change influence phylogenetic diversity may lead to fundamental insights into how ecosystems function. Despite this, scientists still know little about contemporary changes in phylogenetic diversity. Increasingly available data on species occurrence and comprehensive phylogenies for multiple taxonomic groups [23,24] now make it possible to explore how anthropogenic change alters phylogenetic diversity at the global scale.

We provide one of the first comprehensive meta-analyses of changes in taxonomic and phylogenetic $\alpha$ and $\beta$ diversity across multiple taxonomic groups by compiling 162 datasets from across the globe (figure 1). These datasets represent three broad data types based on their sampling methods: land use data ($n = 66$), species list data ($n = 78$) and resample data ($n = 18$). Each dataset consists of a 'historical' state and a 'current' state for at least three locales. The historical state is defined as either an undisturbed habitat along a land use gradient of low to high human impact, current regional species lists excluding non-native species, or an initial sampling time period [25]. This historical state represents ecological assemblages that are less impacted by human actions. The current state is defined as either human-impacted habitat along the land use gradient,

regional species lists that include non-native species and exclude native species that have been recently extirpated, or the second of two sampling periods (greater than 20 years after the first). Thus, the current state reflects recent change that is probably driven by anthropogenic impacts on species composition (see the electronic supplementary material, table S5 for details about each dataset). Our datasets include 32 382 plant species ($n = 53$), 2903 bird species ($n = 54$) and 13 236 fish species ($n = 55$), and span five continents (figure 1).

We address three primary questions. First, how have taxonomic and phylogenetic $\alpha$ diversity been altered by anthropogenic change? Second, what evidence is there for decreasing or increasing spatial $\beta$ diversity in taxonomic or phylogenetic composition? Third, are there associations between changes in $\alpha$ and $\beta$ diversity, and changes in phylogenetic and taxonomic diversity? For different data types and major taxonomic groups, we calculated taxonomic and phylogenetic diversity within a locale ($\alpha$ diversity) and $\beta$ diversity from pairwise differences in taxonomic and phylogenetic composition between locales. We used multiple phylogenetic diversity indices, including those that are independent and non-independent of taxonomic diversity. Next, changes in $\alpha$ and $\beta$ diversity from the historical to the current state were quantified using the log ratio of diversity for each dataset (effect sizes, see Material and methods for details). A strength of our approach is that we analysed original data, rather than extracting published summary statistics; this allowed us to account for sampling variance within each dataset and to directly compare changes in $\alpha$ and $\beta$ diversity within each dataset [13].

## 2. Material and methods

### (a) Data collection

Based on the dataset compiled by Baiser *et al.* [11], we collected additional data following the same protocol. We searched the literature and online databases of plants, birds and fishes that had at least *three* locales and reported the following information.

(i) Temporal changes (greater than 20 years) in species composition at the same locales. For this type of study (hereafter

resample data), we treated the first year as the first time period (dat_1) and the last year as the second time period (dat_2) for most studies. The exceptions were studies with annual long-term sampling [26]. For these studies, we used the average values from the first 5 years and the last 5 years as the first time period (dat_1) and the second time period (dat_2), respectively. We limited our scope to studies that covered more than 20 years to account for potential lag effects in changes in species diversity and composition. Because of our focus on spatial homogenization and thus our requirement of three or more locales in a study, we were unable to use many time series in the BioTIME database [27].

(ii) Changes in species composition along an anthropogenic gradient (e.g. urban to natural, farm to forest). For this type of study (hereafter land use data), we took two approaches to designating the first and second 'time periods'. In the first approach, we used data from the natural end of the gradient (e.g. preserved area, forest, primary vegetation) as the first 'time period' (dat_1) and the developed end (e.g. urban, cropland, pasture, plantation forests) as the second 'time period' (dat_2). We will refer to this approach as the 'land use gradient approach'. For the second approach, we again used data from the natural end of the gradient as the first 'time period' (dat_1). For the second 'time period' (dat_2), we combined data from the natural and developed locales. This second approach takes a landscape approach to assess change in biodiversity patterns where dat_1 is considered a landscape consisting of the 'natural' ecosystem while dat_2 consists of a heterogenous mix of 'natural' and developed locales. We will refer to this approach as the 'landscape approach'.

The majority of datasets for the land use category were from a global database of effects of land use on terrestrial biodiversity (the PREDICTS project) [28]. The 2016 release of this dataset had greater than 3.2 million records of greater than 47 000 species sampled at more than 26 000 locations from 666 studies. We focused only on studies of birds and plants. We limited our scope to studies that compared primary vegetation with at least one of the following land use types: plantation forest, cropland, pasture and urban. We chose these four land use types to compare with primary vegetation because a previous study showed that these human land uses dramatically alter primary vegetation [8]. Studies that had more than one land use type (other than primary vegetation) were split into different datasets. For example, if a study had primary vegetation, cropland and urban as land use types, we split it into two datasets: primary vegetation versus cropland and primary vegetation versus urban. In addition, we removed studies that had less than three locales and 10 species in each of their land use types. Therefore, we only included 71 datasets from 42 studies found in the PREDICTS database in our final analysis.

(iii) Native and non-native species (and extirpated species if available) of the same sub-regions (hereafter termed locales). For this type of study (hereafter species list data), we considered native species as the historical time period (dat_1), and native + non-native − extinct (if available) species as the contemporary time period (dat_2). The majority of datasets for this type of study were from a global database of freshwater fishes [29]. This database grouped 3119 basins by country and biogeographic realm. We removed countries that had less than three basins and 10 non-native records across all basins, which resulted in 51 countries. We treated each country as a separate dataset. We then coupled this global fish database with information about extirpated species for basins [30]. However, extirpations were rare across basins with the average number of extirpated species being less than one [31].

Because we wanted to study changes in $\beta$ diversity and phylogenetic diversity, species composition for these studies was required. We requested raw data for studies that met our criteria but were not publicly available by sending emails to corresponding authors. In total, we collected 189 datasets compiled from 77 published studies and databases of plants, birds and fishes across the world (see the electronic supplementary material, table S5 for details). Twenty-six papers (including data papers with global extent datasets [29,32]) reported more than one dataset, which we analysed separately.

Each dataset consisted of species occurrence information for a specific taxonomic group, represented as two presence–absence-by-locale matrices (one for dat_1 and the other for dat_2). Except for studies from the PREDICTS database, both matrices of the same dataset had the same number of locales. Studies from the PREDICTS database did not always have the same number of locales for each land use type; we thus used the sampling effort corrected measurement for each dataset provided in the database. The locales across studies had different grain sizes, and we categorized those as small (less than 1 km$^2$), moderate (1–100 km$^2$), large (100–1000 km$^2$) and very large (greater than 1000 km$^2$). In addition, datasets had different spatial extents, ranging from regional to global. However, the majority of datasets for which the locales had small grain size were land use or resample datasets, while the majority of datasets with moderate to high grain size were species list datasets (electronic supplementary material, figure S1). For each dataset, we extracted the main drivers of change from the corresponding original paper. We classified all drivers into eight groups: urbanization, grazing, agriculture, invasion, management, climate change, post-disturbance and on-going disturbance. We checked datasets for duplicates and removed locales without any species recorded for all datasets. In the end, we collected information on 32 382 plant species at 2827 locales from 63 datasets, 2903 birds at 2525 locales from 58 datasets and 13 236 fishes at 3029 locales from 68 datasets. We checked and standardized all 48 521 species' names based on the Taxonomic Name Resolution Service [33] using the R package taxize v. 0.9.3 [34]. These species represent about 2.55% of described species in the world [35].

## (b) Phylogenies

For plants, we generated the phylogeny using PHYLOMATIC v. 4.2 [36] based on the synthesis phylogeny zanne2014 [37]. For birds, because we had extinct birds in the historical data period [32], we used the 100 augmented phylogenies constructed by Baiser et al. [38]. For fishes, we generated the phylogeny using PHYLOCOM [39] and its bladj function based on the OPEN TREE OF LIFE [24] and the TimeTree of Life database [40]. See the electronic supplementary material for details of the phylogeny building processes. Phylogenies generated by the OPEN TREE OF LIFE and PHYLOMATIC usually contain multiple polytomies, which are generally at the genus level (near the terminal tips). While not ideal, the OPEN TREE OF LIFE and PHYLOMATIC are virtually the sole operational tools to construct phylogenies without detailed phylogenetic information for thousands of species. A previous simulation study suggests that community level phylogenetic diversity metrics are robust against polytomies at the terminal side [41]. Furthermore, recent research has shown that phylogenetic diversity metrics calculated from phylogenies built with our methods are highly correlated with metrics derived from 'purpose-built' phylogenies based on gene sequences [42]. Therefore, we are confident that the phylogenies are robust in terms of phylogenetic diversity measurements.

## (c) Diversity metrics

Most of our datasets had no information on species abundance, therefore, we converted all datasets into presence/absence data (1 and 0). We then used SR to measure taxonomic $\alpha$ diversity. For $\beta$ diversity, we calculated measures of spatial turnover (pairwise dissimilarity of locales) in species composition for both site-by-species

matrices (dat_1 and dat_2) of each dataset using the Sorensen dissimilarity index. We used pairwise dissimilarity to calculate $\beta$ diversity because this allows us to calculate the variance in dissimilarity and weight datasets inversely by this variance (see details in the Statistical analysis section). Multi-site dissimilarity metrics do not allow us to calculate a variance within studies for use in weighting, because a multi-site metric returns only one value for each site-by-species matrix. We calculated the additive components (nestedness and species turnover) of Sorensen dissimilarity using the R package betapart v. 1.5.0 [43,44]. We then used the turnover component (Sorensen_tur) as our measure of taxonomic $\beta$ diversity because it is independent of, but complementary to, measurements of $\alpha$ diversity [43]; furthermore, the turnover component accounts for greater than 75% of the Sorensen dissimilarity for most of the datasets (electronic supplementary material, figure S2) and gives qualitatively similar results as the Sorensen dissimilarity (electronic supplementary material, table S4).

The large number of phylogenetic diversity metrics can be clustered into three groups that characterize species: richness, divergence and regularity [45]. To get a better understanding of changes in phylogenetic diversity, we used metrics that are both independent and non-independent of taxonomic diversity. For metrics that are non-independent of taxonomic diversity, we used Faith's PD (phylogenetic diversity) [14] to measure phylogenetic $\alpha$ diversity, and the turnover component (pSorensen_tur) of phylogenetic Sorensen dissimilarity [46] to measure phylogenetic $\beta$ diversity. For metrics that are independent of taxonomic diversity, we used phylogenetic species variability (PSV) [47], which is equivalent to mean pairwise distance (MPD) [47,48], to calculate phylogenetic $\alpha$ diversity. PSV quantifies expected mean phylogenetic distance (electronic supplementary material, figure S3); it approaches one when all species in a locale are unrelated and approaches zero when species are closely related. For phylogenetic $\beta$ diversity, we used phylogenetic community dissimilarity (PCD) [49]. PCD is based on PSV and measures pairwise phylogenetic dissimilarity by asking how much of the variance of values of a hypothetical non-selected trait among species in one locale can be predicted by the values of species from another locale. Because PCD is standardized by SR in the species pool, datasets with different species pools cannot be compared directly. We thus used species of both dat_1 and dat_2 as species pools for each dataset. A lower PCD value for dat_2 suggests phylogenetic homogenization while a higher value suggests phylogenetic differentiation. We used the phylogenetic component of PCD (PCDp) as the phylogenetic $\beta$ diversity metric that is independent of taxonomic diversity [49].

### (d) Statistical analysis

Because of differences in sampling methodologies, taxa, spatial extents and grain sizes, we kept datasets separate when calculating diversity. For each dataset, we calculated $\alpha$ diversity (SR, Faith's PD and PSV) for each locale and $\beta$ diversity for all unique combinations of pairs of locales (Sorensen_tur, pSorensen_tur and PCDp) within both matrices. For each $\alpha$ and pairwise $\beta$ diversity index, we calculated the mean value for both matrices ($\overline{X_1}$ and $\overline{X_2}$). For each dataset, we then quantified changes in each diversity index as the log response ratio (LRR) = $\log(\overline{X_2}/\overline{X_1})$. For species list datasets, $X_2$ is the average taxonomic or phylogenetic diversity based on native and non-native species while $X_1$ is based on native species excluding any species that went extinct/extirpated (if any). For resample datasets, $X_2$ is based on the average taxonomic or phylogenetic diversity of the resurveys while $X_1$ is based on the initial survey. For the land use data using the land use gradient approach, $X_2$ is the average taxonomic or phylogenetic diversity of disturbed sites (e.g. urban or farm) while $X_1$ is the average taxonomic or phylogenetic diversity of undisturbed sites (e.g. forest). For the land use data using the 'landscape' approach, $X_1$ is the average taxonomic or phylogenetic diversity

of undisturbed sites but $X_2$ is the average taxonomic or phylogenetic diversity of disturbed and undisturbed sites.

We divided datasets based on taxonomic groups, data types, and continents and then analysed them separately. For each subset of the datasets, to test whether $\alpha$ and $\beta$ diversity differed significantly between dat_1 and dat_2, we used linear mixed models (LMMs) with the LRR as the response variable and the intercept as the only fixed term. To account for the differences in datasets, we included the original study name, the data type (i.e. land use, species list and resample; when analysed by taxonomic groups), taxonomic group (i.e. bird, fish and plant; when analysed by data types), grain size and main driver as random terms. We included these variables as random terms instead of fixed terms because we were mainly interested in quantifying the changes in (not the predictors of) LRR in this study. Taxonomic group and data type both only have three levels; however we still included them as random terms because doing so can make the model easier to interpret while not performing any worse than a classical regression [50]. A random intercept for study accounts for the hierarchical structure given that multiple datasets are from the same study.

All LMMs were fitted with the R package lme4 v. 1.1–17 [51]. After fitting the full model, we selected the random terms by likelihood ratio tests using the R package lmerTest v. 2.0–36 [52]. The 95% confidence interval (CI) of the intercept was assessed by computing scaled Wald statistics, which were treated as following an approximate $F$ distribution [53]. For all diversity metrics, significant positive LRR suggests higher diversity in dat_2 and *vice versa*.

For all LMMs, we weighted each dataset inversely by the sampling variance of LRR. Specifically, for datasets from land use studies, we quantified the LRR sampling variance as

$$\hat{\sigma}_{\mathrm{LRR}}^2 = \frac{\mathrm{Var}_{X_1}}{n_1 \overline{X_1}^2} + \frac{\mathrm{Var}_{X_2}}{n_2 \overline{X_2}^2}.$$

For the other two study types (i.e. species list data and resample data), measures of diversity from dat_1 were correlated with those from dat_2, which can influence the accuracy of the variance estimate. Therefore, we used a recent method proposed by Lajeunesse [54] to account for correlations. For these datasets, we quantified the LRR sampling variance as

$$\hat{\sigma}_{\mathrm{LRR}}^{2'} = \frac{\mathrm{Var}_{X_1}}{n_1 \overline{X_1}^2} + \frac{\mathrm{Var}_{X_2}}{n_2 \overline{X_2}^2} - \frac{2\rho \mathrm{Sd}_{X_1}\mathrm{Sd}_{X_2}}{\overline{X_1}\,\overline{X_2}\sqrt{n_1 n_2}}.$$

To better understand why diversity changed, for each dataset, we identified the species lost (species that are unique in dat_1) and the species gained (species that are unique in dat_2) for each pair of sites (for species list data and resample data) or for all possible combinations of sites within each land use category for land use data. We then compared the average number of species gained and lost, the average number of sites in which gains and losses were observed, and the mean pairwise phylogenetic distances between shared and gained species (MPDsg) and shared and lost species (MPDsl) with paired $t$-tests.

### 3. Results

Taxonomic and phylogenetic $\alpha$ diversity changed consistently across different taxonomic groups. Fishes, birds and plants all demonstrated marked increases in SR (10.62%, 4.69% and 6.18%, respectively), and Faith's PD (8.21%, 2.84% and 3.37%, respectively) with fishes showing significant increases (figure 2a,c). However, after accounting for the changes in SR, all three taxa showed decreases in phylogenetic $\alpha$ diversity (fishes = −0.11%, plants = −0.77% and birds = −2.08%, measured by PSV), and changes for fishes and plants were statistically significant (figure 2c).

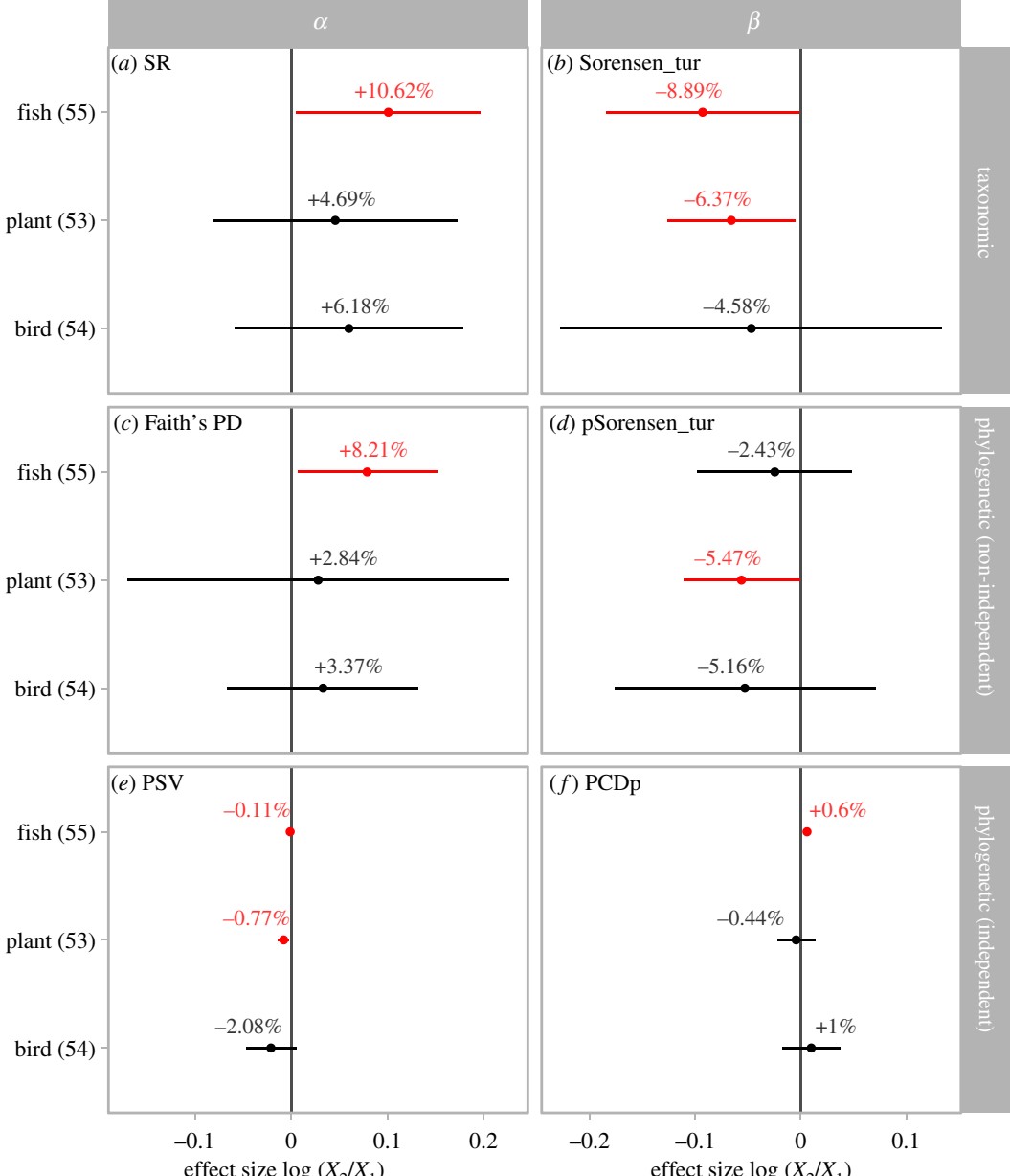

**Figure 2.** Changes in biodiversity of different taxonomic groups. (*a*) Changes in taxonomic $\alpha$ diversity measured as species richness; (*b*) pairwise taxonomic $\beta$ diversity measured as the turnover component of Sorensen dissimilarity; (*c*) phylogenetic $\alpha$ diversity measured as Faith's PD; (*d*) pairwise phylogenetic $\beta$ diversity measured as the turnover component of phylogenetic Sorensen dissimilarity; (*e*) phylogenetic $\alpha$ diversity measured as PSV; and (*f*) pairwise phylogenetic $\beta$ diversity measured as the phylogenetic component of phylogenetic community dissimilarity. Error bars show mean and 95% confidence intervals based on linear mixed models, and are red if the confidence interval does not include zero. Positive effect sizes indicate increases in diversity. The numbers in parentheses denote the number of studies. (Online version in colour.)

Changes in taxonomic and phylogenetic $\alpha$ diversity showed marked differences across data types. For species list data; both SR and Faith's PD increased (by 6.93% (95% CI: +3.69% to +10.28%) and 4.59% (95% CI: +2.53% to 6.61%), respectively; figure 3*a*,*c*), while PSV decreased by 0.25% (95% CI: −0.49% to −0.01%; figure 3*e*). For resample data; SR, Faith's PD and PSV all increased (by 17.09%, 15.18% and 1.19%, respectively) but changes were not significant. For land use data; SR, Faith's PD and PSV all decreased significantly when using the land use gradient approach (by 10.22% (95% CI: −18.86% to −0.67%), 12.29% (95% CI: −19.10% to −4.88%) and 3.31% (95% CI: −5.32% to −1.27%), respectively; figure 3*a*,*c*,*e*). When using the land-scape approach; SR, Faith's PD and PSV still decreased but only significantly for PSV (by 1.04% (95% CI: −5.86% to

4.03%), 2.35% (95% CI: −6.76% to 2.28%) and 1.05% (95% CI: −1.70% to −0.39%), respectively; electronic supplementary material table S2).

Taxonomic $\beta$ diversity decreased for all taxonomic groups (figure 2*b*): −8.89% (95% CI: −16.86% to −0.16%) for fishes, −6.37% (95% CI: −11.87% to −0.53%) for plants, and −4.58% for birds. Taxonomic $\beta$ diversity also decreased for all data types (figure 3*b*): −6.24% in land use data using the land use gradient approach, −4.17% in species list data and −9.6% (95% CI: −17.37% to −1.1%) in resample data. When measured with the turnover component of phylogenetic Sorensen dissimilarity, phylogenetic $\beta$ diversity also decreased for all taxonomic groups and data types, mirroring those changes in taxonomic $\beta$ diversity (figure 2*b* versus *d*; figure 3*b* versus *d*). The only exception is for the land use data using the

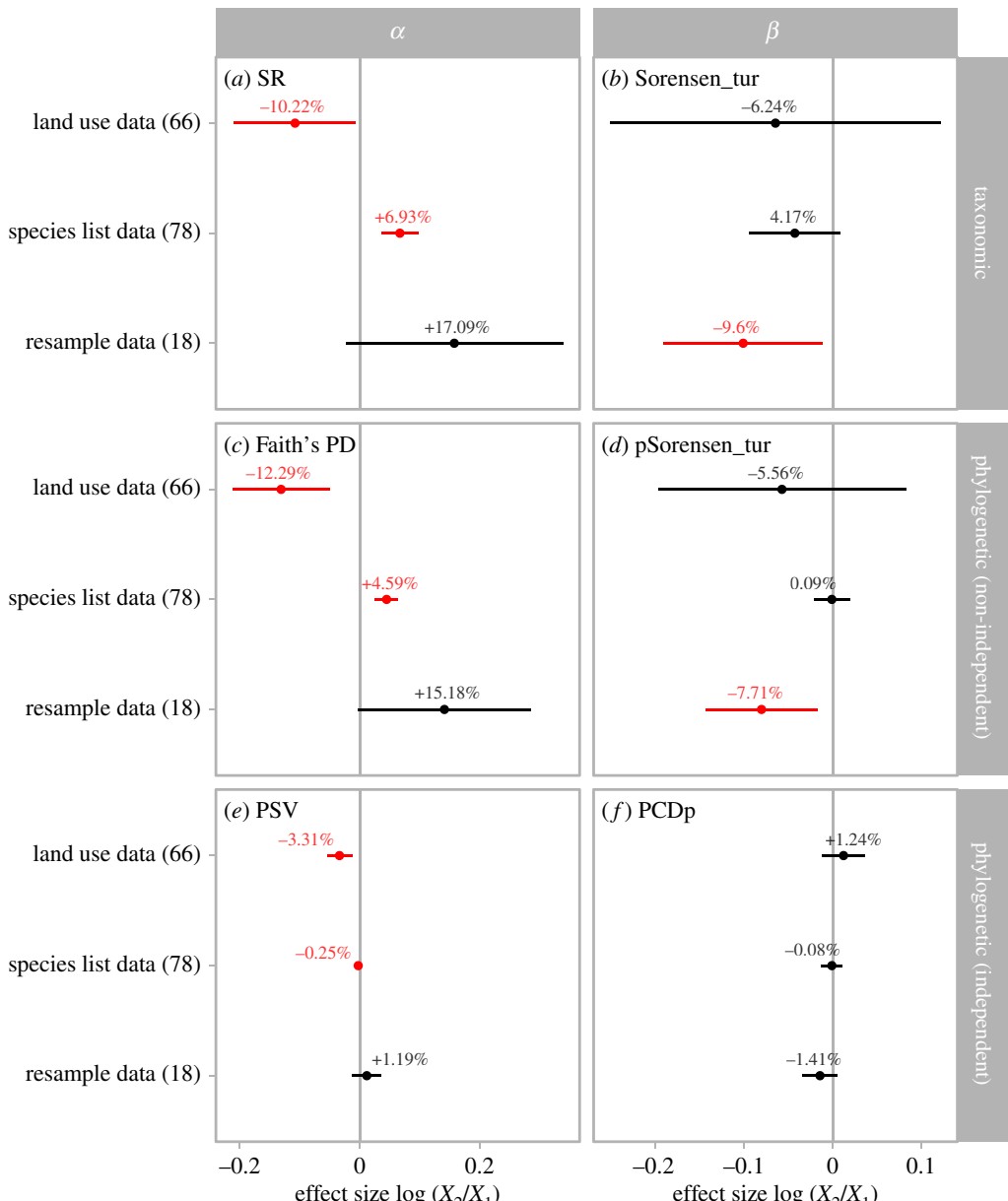

**Figure 3.** Changes in biodiversity of different data types. (*a*) Changes in taxonomic $\alpha$ diversity measured as species richness; (*b*) pairwise taxonomic $\beta$ diversity measured as the turnover component of Sorensen dissimilarity; (*c*) phylogenetic $\alpha$ diversity measured as Faith's PD; (*d*) pairwise phylogenetic $\beta$ diversity measured as the turnover component of phylogenetic Sorensen dissimilarity; (*e*) phylogenetic $\alpha$ diversity measured as PSV; and (*f*) pairwise phylogenetic $\beta$ diversity measured as the phylogenetic component of phylogenetic community dissimilarity. Error bars show mean and 95% confidence intervals based on linear mixed models, and are red if the confidence interval does not include zero. Positive effect sizes indicate increases in diversity. The numbers in parentheses denote the number of studies. For land use data, results here were derived from the land use gradient approach; see the electronic supplementary material, table S2 for results derived from the landscape approach. (Online version in colour.)

landscape approach (electronic supplementary material, table S2), which suggested a marginally significant increase in taxonomic $\beta$ diversity (by 7.88% with 95% CI between 0.8% and 15.45%) and a non-significant increase in phylogenetic $\beta$ diversity (by 5.88% with 95% CI between −0.96% and 13.21%). However, after accounting for changes in taxonomic $\beta$ diversity by using the PCDp metric, changes in phylogenetic $\beta$ diversity across taxonomic groups and data types were inconsistent with no significant changes with the exception of fishes (figures 3*d* and 2*d*).

Overall, $\alpha$ diversity and $\beta$ diversity tended to change in different directions (figures 2 and 3). Furthermore, when taxonomic diversity and phylogenetic diversity were calculated independent of SR, substantial increases and decreases in taxonomic $\alpha$ and $\beta$ diversity did not lead to corresponding increases

and decreases in phylogenetic $\alpha$ and $\beta$ diversity. In fact, changes in phylogenetic $\alpha$ and $\beta$ diversity were generally in the opposite direction from taxonomic $\alpha$ and $\beta$ diversity (figure 2*a* versus *e*, *b* versus *f*). One possible reason for the disconnection between phylogenetic and taxonomic change is that the phylogenetic distances between gained species and shared species were not statistically different from the phylogenetic distances between lost and shared species for all taxonomic groups (all $p > 0.05$, paired *t*-tests, electronic supplementary material, figure S8) and for all data types except land use data (0.015 versus 0.025, d.f. = 65, $p = 0.021$, paired *t*-tests, electronic supplementary material figure S9).

The average number and occupancy of species gained and lost provide insight into potential mechanisms for changes in $\alpha$ and $\beta$ diversity (electronic supplementary material, figures S6

and S7). For fishes, because most of the datasets are derived from lists of native and non-native species (electronic supplementary material, figure S1a), the locale-level average number of species added across all studies (3.5) was higher than the number of species extirpated (0.4, d.f. = 54, $p < 0.001$, paired $t$-tests). Fish species that established in new locales tended to occupy a greater average proportion of locales than species that were lost (0.215 versus 0.031, d.f. = 54, $p < 0.001$, paired $t$-tests). For plants, we also found that more species tended to establish in new locales than species that were lost across all datasets (58.9 versus 31.8, d.f. = 52, $p = 0.04$, paired $t$-tests), and they occupied more sites than lost plants (0.157 versus 0.097, d.f. = 52, $p < 0.001$, paired $t$-tests). However, birds showed the opposite pattern: on average 9.1 species were added while 11.8 species were lost within local assemblages (d.f. = 53, $p = 0.044$, paired $t$-tests). The extirpated birds tended to occupy more locales than gained species (mean site occupancy 0.211 versus 0.198), but this difference was not significant. For fishes and plants, the fact that newly established species were widespread and extirpated species were more locally distributed (electronic supplementary material, figure S6) probably led to the observed significant taxonomic homogenization for these taxa (figure 2c). Similar patterns were found in species list data and resample data (electronic supplementary material, figure S7), which probably led to the observed taxonomic homogenization patterns for these data types (figure 3c).

## 4. Discussion

Recent global syntheses on contemporary changes in biodiversity have largely focused on taxonomic $\alpha$ diversity, typically SR at local scales, and found divergent results. Studies synthesizing temporal data found little evidence [4,5] for overall declines in $\alpha$ diversity while those that focused on land use change found declines in $\alpha$ diversity [8]. Our results confirm that biodiversity information derived using different sampling methods differ in the direction of SR change. Both species list and resample data experienced increases in taxonomic $\alpha$ diversity over time. However, land use data—those that compare diversity across locales that differ in the degree of human impact—show large declines in taxonomic $\alpha$ diversity, in agreement with a recent study on the effects of land use changes on biodiversity more generally [8]. Despite different trends in SR, nearly all data types showed decreased taxonomic $\beta$ diversity, suggesting that changes in $\beta$ diversity can be decoupled from changes in $\alpha$ diversity. The exception to the general trend of decreasing $\beta$ diversity occurred when we looked at land use data from a 'landscape' perspective. In this scenario, we assume that time period one has a homogeneous landscape of undisturbed habitat (e.g. forest) and that during time period two, the landscape is a mix of undisturbed and disturbed habitat (e.g. forest and suburban). Based on this assumption, it is no surprise that $\beta$ diversity increases in this scenario because a new habitat type (i.e. disturbed habitat) and its constituent species are added to the landscape. However, this assumption probably does not hold in most landscapes which are a heterogenous mix of habitats even if they are not anthropogenically impacted. Nevertheless, such an approach at the landscape scale is useful to understanding changing biodiversity spatial patterns [55].

All three taxonomic groups investigated showed increasing taxonomic $\alpha$ diversity and decreasing taxonomic $\beta$ diversity consistently when combining all data types together, with significant changes in $\alpha$ and $\beta$ diversity of fish communities and $\beta$ diversity of plant communities. The majority of fish community data came from species list data (electronic supplementary material, figure S1), which showed increasing taxonomic $\alpha$ diversity and decreasing taxonomic $\beta$ diversity (figure 2). Therefore, the observed changes in taxonomic diversity of fish communities were not surprising. For plant and bird communities, more than half of the datasets were from land use data (electronic supplementary material, figure S1), which showed decreasing taxonomic $\alpha$ and $\beta$ diversity (figure 2). However, taxonomic $\alpha$ diversity of both plants and birds did not show a declining trend. These results suggest that multiple types of diversity information should be used simultaneously to gain a more complete picture of biodiversity change [6,7]. The challenge now is to determine if and how one should integrate results from different types of datasets to draw general conclusions. Cardinale et al. [7] suggested that results from different data types can be weighted by their spatial coverage. However, such information may not always be available, which is the case for the datasets we collected here.

Changes in phylogenetic diversity, when quantified with metrics that are non-independent with taxonomic diversity (Faith's PD and phylogenetic Sorensen dissimilarity), showed the same trends as taxonomic diversity. We found increasing phylogenetic $\alpha$ diversity but decreasing phylogenetic $\beta$ diversity for all taxonomic groups and data types, except phylogenetic $\alpha$ diversity of land use data, which declined. However, when quantified with metrics that are independent of taxonomic diversity (PSV and PCDp), changes of phylogenetic diversity do not necessarily follow changes in taxonomic diversity. For example, all taxonomic groups and species list data showed increasing SR but decreasing phylogenetic $\alpha$ diversity (PSV). Therefore, species invasion may reduce average phylogenetic distance among species in the community despite its potential positive effects on taxonomic diversity at large scales [38]. It is also important to note that land use changes reduced both SR and PSV. Resample data showed no overall decline in phylogenetic $\alpha$ diversity (Faith's PD and PSV), which agrees with predictions of phylogenetic diversity for Europe under future climate change scenarios [56]. Differences in results based on the dependence of phylogenetic diversity measures on SR suggests that knowledge of which class of metrics better reflects ecosystem integrity is important for interpreting patterns of phylogenetic change.

To the best of our knowledge, this study represents the most comprehensive assessment of biotic homogenization. The consistent trend of decreasing taxonomic and phylogenetic (when measured with phylogenetic Sorensen dissimilarity) $\beta$ diversity across taxonomic groups, sampling methods, and continents (electronic supplementary material, figure S4) provides strong evidence for widespread biotic homogenization. This result is in line with other recent studies on taxonomic [11,13] and phylogenetic homogenization [56,57], providing compelling evidence that multiple aspects of the Earth's biodiversity have been mixed by the anthropogenic blender. Given the important influences of biotic homogenization on ecological and evolutionary processes [58], and ecosystem multi-functionality [59], future studies should more clearly focus on the consequences of biotic homogenization on ecosystem services.

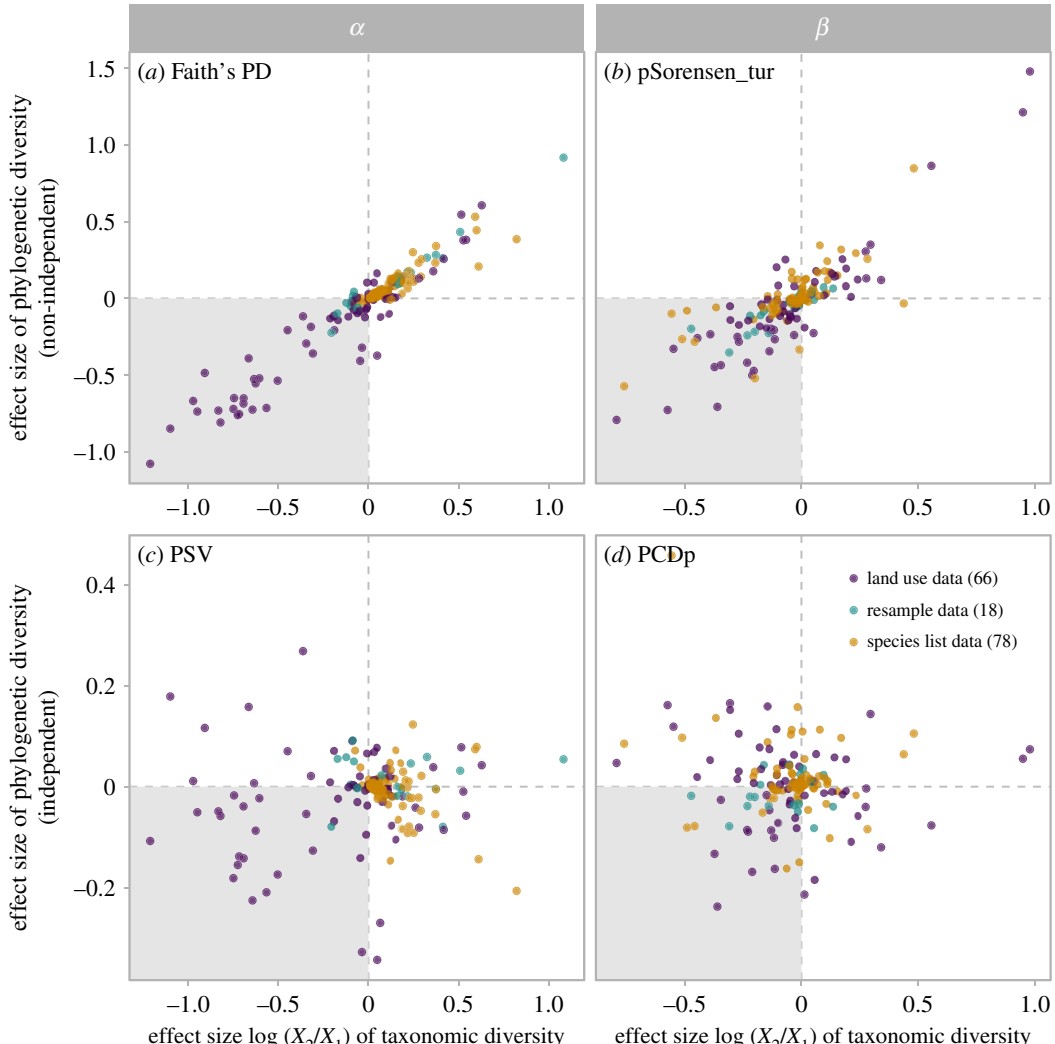

**Figure 4.** Relationships between effect sizes of taxonomic diversity and phylogenetic diversity. (*a*) Phylogenetic α diversity measured as Faith's PD; (*b*) pairwise phylogenetic β diversity measured as the turnover component of phylogenetic Sorensen dissimilarity; (*c*) phylogenetic α diversity measured as PSV; and (*d*) pairwise phylogenetic β diversity measured as the phylogenetic component of phylogenetic community dissimilarity. The positive effect size suggests increases in diversity. Therefore, the grey area indicates decreases in both taxonomic diversity and phylogenetic diversity. Each dot represents the observed effect size for a dataset. Land use datasets consistently have reduced taxonomic α diversity. Most points in (*b*) and (*d*) are at the left side of the vertical line, suggesting taxonomic homogenization, a pattern confirmed by weighted linear mixed models. For land use data, results here were derived from the land use gradient approach. (Online version in colour.)

Although our data include tens of thousands of species from different taxonomic groups across the globe, we must acknowledge several caveats in our analysis. First, our data are geographically biased toward certain regions of the world, as are most biodiversity data (figure 1). In regions with no or little data, it is currently impossible for us to estimate biodiversity changes. Recent efforts to fill in such geographical biodiversity data gaps include the use of citizen scientists [60], advances in technology (e.g. high-throughput sequencing [61], remote sensing [62]), and using museum and herbarium specimens and local species inventories to develop regional floras and faunas [63,64]. Filling in these geographical biodiversity gaps remains a top priority in quantifying global patterns in biodiversity change. Further, the different types of data we used each have inherent biases. For example, the resample data probably have different baselines, which may affect the biodiversity trends [6,7]; species list data may not have accurate estimates of extirpated species given the challenge associated with confirmation of population extinctions,

resulting in bias towards increasing biodiversity (especially SR); and land use data uses space-for-time substitution to reflect the influence of people on local biodiversity, and have different prevalence, which may also give biased results [65]. Finally, multiple drivers of environmental changes generally happen simultaneously and probably interact with each other in the real world.

Based on these caveats, our results should be interpreted only in the context of our database. Nonetheless, the size of the database makes our results informative about a large number of species in diverse regions of the world. In addition, the decoupling of α and β diversity, as well as taxonomic and phylogenetic diversity (when measured with metrics that are independent of taxonomic diversity), holds across different taxonomic groups and different data types, suggesting a strongly coherent global pattern. While insights into changing biodiversity patterns can be gained through the synthesis of heterogeneous data [66], critiques [6,7] should be used to move the study of biodiversity change forward.

## 5. Conclusion

We derive three key insights from our global synthesis. First, changes within assemblages ($\alpha$ diversity) do not reflect spatial changes ($\beta$ diversity). We show that taxonomic $\beta$ diversity decreased, suggesting that species composition among assemblages is becoming more homogeneous despite a lack of decline in taxonomic $\alpha$ diversity in most situations. Second, changes in taxonomic diversity do not necessarily reflect changes in phylogenetic diversity, suggesting the importance of quantifying multiple facets of biodiversity when assessing the impact of anthropogenic activities on biodiversity. Third, assemblages show tremendous variation in diversity change, both in terms of taxonomy and evolutionary relationships (figures 2–4; electronic supplementary material, figures S5–S9). We suggest that future research into responses of biodiversity to global change should use a variety of metrics that reflect multiple facets of diversity at different spatial scales [67]. Such investigations may provide an enhanced understanding of the current biodiversity crisis and will be essential to safeguarding ecosystems in the Anthropocene.

Data accessibility. Data from the PREDICTS dataset and the global fish dataset were publicly available from their own websites. Other datasets were shared by authors for use in this publication only and as a result, we do not have the permission to make them publicly available. Please contact the authors directly for inquiries about these datasets. Phylogenies and calculated taxonomic and phylogenetic diversity values for each dataset are archived in the Dryad Digital Repository: https://dx.doi.org/10.5061/dryad. 3j9kd51fc [68].

Authors' contributions. D.L. and B.B. planned the project. D.L. conceived and ran analyses and wrote the first draft of the manuscript. All authors contributed to interpreting the results, and the writing and editing of the manuscript.

Competing interests. The authors declare no conflict of interest.

Funding. This research was funded in part by NSF ABI no. 1458034 to B.B., the K-G Research fund of Bryn Mawr College to S.R. and the UW Mason H. Keeler Endowed Professorship to J.D.O.

Acknowledgements. We thank many people and their co-authors (see the electronic supplementary material for all contributors) for sharing their data, without which this paper would not have been possible. We also thank Anthony R. Ives and two anonymous reviewers for critical and helpful comments on the manuscript.

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
