## [Reviewer comments · Proceedings of the Royal Society B: Biological Sciences]

Review History

RSPB-2019-2404.R0 (Original submission)

Review form: Reviewer 1

Recommendation

Major revision is needed (please make suggestions in comments)

Scientific importance: Is the manuscript an original and important contribution to its field?

Good

General interest: Is the paper of sufficient general interest?

Excellent

Quality of the paper: Is the overall quality of the paper suitable?

Excellent

Is the length of the paper justified?

Yes

Should the paper be seen by a specialist statistical reviewer?

Yes

Do you have any concerns about statistical analyses in this paper? If so, please specify them explicitly in your report.

Yes

It is a condition of publication that authors make their supporting data, code and materials available - either as supplementary material or hosted in an external repository. Please rate, if applicable, the supporting data on the following criteria.

Is it accessible?

No

Is it clear?

Yes

Is it adequate?

N/A

Do you have any ethical concerns with this paper?

No

Comments to the Author

This is an interesting and well-written manuscript putting together several threads from the literature concerning biodiversity change: how results vary among data sources, how alpha and beta diversity change simultaneously, and whether phylogenetic diversity changes mirror those found for taxonomic diversity. The authors use an impressive combination of data sets for multiple taxa and geographic regions to tackle these questions. The analyses include quite a few steps involving data subsetting, choices of metrics, and model structure; most justifications seems fine, although there are some areas where I think important conclusions depend on questionable decisions (described below). The results to a considerable extent confirm what is already known: land use can cause alpha diversity declines (as revealed by space-for-time datasets); other kinds of changes influencing biodiversity do not cause declines on average (revealed in other data sets); and biotic homogenization is common. The phylogenetic results are definitely a new addition, and having all of these things in one place is useful as well. Overall, I think this study is likely to make an important contribution to the literature on this topic.

Note: I am not knowledgeable enough of phylogenetic methods to comment on that; I also can't keep track of all the dos and don'ts of statistical practice, although the use of random effects for categories that sure seem like fixed effects, or with very small numbers of categories, seems to violate common advice on best practices.

(1) Land use and biotic homogenization. For the Resample and Species List data sets, I think the inferences about biotic homogenization are valid, in that particular set of places become more similar over time. For the Land Use data, I think there is a flaw in the logic. When one converts natural habitats to urban areas or crops (for example), it is not the case that a landscape is now entirely urban or agricultural. Rather, the modified landscape is a mix of these different land cover types. So, asking how land use has altered beta diversity requires that dat_2 (the data matrix representing "after") include not only those plots under a given land use, but all plots. If the species composition of crop fields is different from that in natural vegetation, then one may well expect that beta diversity will have done up at the landscape scale, although the outcome will depend on variation both within and among land cover types. In short, if you think about the places where these land use studies are done, people can create land cover types for which replicate plots are compositionally very similar to one another, while at the same time creating a landscape that is not more homogenous but just the opposite. Looking at things both ways would be quite straightforward with the data at hand, and provide a richer picture of how land

use has influenced beta diversity.

(2a) Metrics of phylogenetic diversity. I think it's great that the authors recognize and report analyses of phylogenetic diversity both "overall" (i.e., metrics correlated with richness), and a component that is independent of richness. However, the decision to focus the main analyses on richness-independent measures has important consequences for the validity of arguments linking PD to things like resilience, stability, and ecosystem function (lines 54, 377), and even for the generality of any conclusion concerning PD in general. For example, a typical situation in many Species List data sets seems to be few extinctions and a bunch of additions; for sake of argument, imagine that there are only additions. With the addition of new species, the total amount of the tree of life represented can only go up. But if those new species are closely related to residents, something like MPD can go down. I know of no evidence indicating that a change in MPD via this pathway will compromise the stability or functioning of an ecosystem. The kinds of metrics that are correlated with taxonomic richness are at least as likely, and perhaps even more likely, to predict altered ecosystem properties. And I think it's misleading to say in the abstract that "phylogenetic alpha diversity declines for all taxonomic groups", even with the particular metric in parentheses. The truth is that total PD did just the opposite. The fact that total PD and richness are correlated has no bearing on the question of whether total PD or MPD are more or less relevant when interested either in ecosystem function or the conversation of biodiversity more generally. Reporting both sets of results in the main figures and discussing them in the text with appropriate precision and nuance would be easy to do.

(2b) Line 395: Related to the last point, I don't see how these results demonstrate the importance of using richness-independent measures of PD. Just because something different was found doesn't make these any more appropriate for a particular objective. They demonstrate that different metrics behave in different ways, not that it's more important to use one that diverges from taxonomic results.

(3) Effect sizes. The authors stress the importance of results being significantly different from zero, and much less so the effect sizes. But some effect sizes seem really tiny, especially for phylogenetic results. People have asked how phylogenetic diversity influences things like ecosystem function. Do changes of <1% matter in any way that's been measured? Is there any plausible scenario in which a change of <1% might matter for conservation or anything else? For other metrics, changes are on the order of 5-10%, which is larger, but again - do changes in richness of this magnitude matter in studies that have looked at the *effects* of richness? I think the answer is 'no'. So while these changes are interesting, and maybe larger changes are coming, the results as they are seem unlikely to be indicative of other consequences in communities or ecosystems. The manuscript suggests otherwise.

(4) Line 149: I'm not sure what's mean by the "sample effort corrected measurement" here.

(5) Line 157: "management" is pretty vague, and could presumably include something like controlled grazing.

(6) Line 202: Were species removed also from the taxonomic analyses? The justification for not using them in phylogenetic analyses is obvious, but there's no reason to eliminate them from analyses where all necessary information is present.

(7) Line 245: Do you mean `dat_2`? If it's a dissimilarity, then it would be a lower value in `dat_2` that indicates homogenization. Although the description makes it sound like a similarity. Something here should be clarified.

(8) Line 262: Log ratios are commonly used for variables with no upper bound, but it seems unusual for indices of beta diversity. Effectively it's a proportional change in a number that itself is bounded between zero and one. I don't have anything specific to suggest here apart from thinking about whether a log ratio or a difference makes more ecological sense. A statistical

expert might have advice.

(9) Line 362: I don't see any way in which these new results "provide an explanation" for earlier results. They simply reinforce what was already found, without digging deeper as to why. That said, the most parsimonious explanation is that one type of data focuses 100% on land-use impacts, while the other type of data focuses on temporal changes that could have occurred for any one of many reasons. And that we knew already anyway.

Review form: Reviewer 2

Recommendation

Major revision is needed (please make suggestions in comments)

Scientific importance: Is the manuscript an original and important contribution to its field?

Good

General interest: Is the paper of sufficient general interest?

Good

Quality of the paper: Is the overall quality of the paper suitable?

Excellent

Is the length of the paper justified?

Yes

Should the paper be seen by a specialist statistical reviewer?

No

Do you have any concerns about statistical analyses in this paper? If so, please specify them explicitly in your report.

No

It is a condition of publication that authors make their supporting data, code and materials available - either as supplementary material or hosted in an external repository. Please rate, if applicable, the supporting data on the following criteria.

Is it accessible?

No

Is it clear?

Yes

Is it adequate?

No

Do you have any ethical concerns with this paper?

No

Comments to the Author

This paper aims to assess how biodiversity is changing at various spatial scales around the globe. The authors use existing data on birds, fish and plants to assess changes in α and β diversity. They used measures of species richness as well as phylogenetic measures of diversity (independent of richness) to investigate how multiple aspects of diversity are changing. The

datasets that were used in the paper could be of 3 forms: 1) the data could include land-use conversion studies where species richness varied as a function of land-use change, 2) the data could include contemporary species list data where knowledge is available on which species are non-native (i.e. have arrived recently), and others which have gone extinct, and 3) time-series data.

I thought the paper was well-written and the data analysis and figures were appropriate. I have two main, fairly major concerns, and other smaller critiques. I hope that the authors can address the major concerns with either more discussion or further analysis of data.

My first major concern is that there have been a number of studies recently investigating how patterns of diversity are changing (or not), and yet none of them (including this paper) overcome the most limiting factor of understanding how biodiversity is changing across the globe: namely huge spatial sampling bias, and general lack of representation of changes in biodiversity outside of North America, central Europe and a few other exceptional locations (e.g. South Africa, Argentina, and the West Indies). This is well-represented in their Figure 1. Much of the globe is not even represented by a single study, and a large fraction of the terrestrial land surface is represented by a 1-2 datasets. The poor sample size across the globe and hugely biased spatial sampling limits any general statement about what is happening to biodiversity. I think that much more emphasis should be placed on this result: we know very little (in some cases literally nothing) about what is happening to biodiversity. I think the authors should highlight the fact that the general data quantity and distribution is poor, with a greater emphasis placed on the fact that they cannot get estimates, nor confidence intervals around estimates of biodiversity change in many regions of the globe. I would also like to know what the authors recommend to alleviate the situation. How could biodiversity estimates could be generated in regions where sampling programs are not under way and likely won't be initiated in the near future?

Secondly, the three different ways of analyzing biodiversity data appear to have different assumptions and limitations and it would be nice if there were a bit more discussion or perhaps analysis of how this might affect interpretation. The time-series analysis is the least problematic, because presumably estimates of biodiversity are generated in the same way over time. However, I'd be curious to know whether this type of monitoring is done more frequently in pristine or restored sites, rather than in sites undergoing major land-use transitions. If so, then this may bias estimates from this type of data to show positive trends in diversity (as seen in Figure 3). Alternatively, how are extinctions detected in the Species List data? If extinctions require a higher burden of proof (lack of observation for a certain time period), than invasions, this may also bias analysis towards estimates of biodiversity increase. Lastly, the land-use conversions included in the paper only include conversions from a relatively natural to a human-impacted state. This may bias estimates towards findings of biodiversity losses. However, this type of analysis should be standardized or weighted by the prevalence of particular land-use transitions observed within a biome, bioclimatic region, or across the globe. This type of analysis would then not bias our view of biodiversity change simply due to a focus on particular land-use transitions of interest.

Specific minor concerns.

P. 8, L. 133: Doesn't removing countries with <10 non-native records across basins bias your analysis towards observations of increasing diversity?

P. 9, L. 170: Can you familiarize us with the assumptions behind this method of inserting extinct birds into the stem preceding the most recent common ancestor?

P. 10, L. 192: How can this decision to equally space nodes be justified? What impact might it have?

P. 11, L. 202: Did this more strongly affect dat_1 or dat_2? Or were removals randomly or equally distributed between both datasets?

General question about the results: Can you confirm that all sites/studies were investigating locations where human influence was present. i.e. not pristine or restored sites? How representative are the land-use conversion types as a percentage of the types of conversion that we see across the globe?

It's disconcerting that the different data types produce different insights. It suggests that they represent different regions or are differentially reflective of different types of biodiversity change.

P. 16, L. 327: Please spell out the difference between extant and established species.

P. 17, L. 334: Assumes that non-native species were not there 20 years or more ago. Some may be non-native but already well-naturalized.

P. 18, L. 366: Why wouldn't such information be reliable?

P. 18, L. 370: This is interesting, and likely robust.

P. 19, L. 376: The link between phylogenetic and functional diversity is not universal and therefore shouldn't be used without support.

P. 20, L. 400: Why didn't you do this? Or at least present the data on how spatially biased the different types of data are? Land-use: how well do these land-use conversions reflect what's happening globally? Species lists: Are species generally gained only because extinctions are hard to document? Time-series: Are these spatially biased? Occurring in pristine or restored, or largely protected environments?

Decision letter (RSPB-2019-2404.R0)

13-Dec-2019

Dear Dr Li,

I am writing to inform you that your manuscript RSPB-2019-2404 entitled "Changes in Taxonomic and Phylogenetic Diversity in the Anthropocene" has, in its current form, been rejected for publication in Proceedings B.

This action has been taken on the advice of referees, who have recommended that substantial revisions are necessary. We are all agreed that this is a very interesting topic and an impressive data set, but the reviewers have identified several major issues. With this in mind we would be happy to consider a resubmission, provided the comments of the referees are fully addressed. However please note that this is not a provisional acceptance.

Please also check the accessibility and adequacy of your archived data files, as one of the reviewers commented that these were not sufficient.

Yours sincerely,
Professor Loeske Kruuk
Editor
mailto: proceedingsb@royalsociety.org

Associate Editor
Board Member: 1

Comments to Author:

The authors have put together an impressive dataset and applied some interesting analyses, including phylogenetic ones that make this a paper with lots of potential. The reviewers have suggested major revisions and made some important points which may require some re-analysis. In particular the authors should reconsider how their land-use conversions are incorporated into the analyses as well as their conclusions from the phylogenetic diversity metrics. Other important changes have been recommended as well and I look forward to seeing a revised version of the manuscript.

Reviewer(s)' Comments to Author:

Referee: 1

Comments to the Author(s)

This is an interesting and well-written manuscript putting together several threads from the literature concerning biodiversity change: how results vary among data sources, how alpha and beta diversity change simultaneously, and whether phylogenetic diversity changes mirror those found for taxonomic diversity. The authors use an impressive combination of data sets for multiple taxa and geographic regions to tackle these questions. The analyses include quite a few steps involving data subsetting, choices of metrics, and model structure; most justifications seem fine, although there are some areas where I think important conclusions depend on questionable decisions (described below). The results to a considerable extent confirm what is already known: land use can cause alpha diversity declines (as revealed by space-for-time datasets); other kinds of changes influencing biodiversity do not cause declines on average (revealed in other data sets); and biotic homogenization is common. The phylogenetic results are definitely a new addition, and having all of these things in one place is useful as well. Overall, I think this study is likely to make an important contribution to the literature on this topic.

Note: I am not knowledgeable enough of phylogenetic methods to comment on that; I also can't keep track of all the dos and don'ts of statistical practice, although the use of random effects for categories that sure seem like fixed effects, or with very small numbers of categories, seems to violate common advice on best practices.

(1) Land use and biotic homogenization. For the Resample and Species List data sets, I think the inferences about biotic homogenization are valid, in that particular set of places become more similar over time. For the Land Use data, I think there is a flaw in the logic. When one converts natural habitats to urban areas or crops (for example), it is not the case that a landscape is now entirely urban or agricultural. Rather, the modified landscape is a mix of these different land cover types. So, asking how land use has altered beta diversity requires that dat_2 (the data matrix representing "after") include not only those plots under a given land use, but all plots. If the species composition of crop fields is different from that in natural vegetation, then one may well expect that beta diversity will have gone up at the landscape scale, although the outcome will depend on variation both within and among land cover types. In short, if you think about

the places where these land use studies are done, people can create land cover types for which replicate plots are compositionally very similar to one another, while at the same time creating a landscape that is not more homogenous but just the opposite. Looking at things both ways would be quite straightforward with the data at hand, and provide a richer picture of how land use has influenced beta diversity.

(2a) Metrics of phylogenetic diversity. I think it's great that the authors recognize and report analyses of phylogenetic diversity both "overall" (i.e., metrics correlated with richness), and a component that is independent of richness. However, the decision to focus the main analyses on richness-independent measures has important consequences for the validity of arguments linking PD to things like resilience, stability, and ecosystem function (lines 54, 377), and even for the generality of any conclusion concerning PD in general. For example, a typical situation in many Species List data sets seems to be few extinctions and a bunch of additions; for sake of argument, imagine that there are only additions. With the addition of new species, the total amount of the tree of life represented can only go up. But if those new species are closely related to residents, something like MPD can go down. I know of no evidence indicating that a change in MPD via this pathway will compromise the stability or functioning of an ecosystem. The kinds of metrics that are correlated with taxonomic richness are at least as likely, and perhaps even more likely, to predict altered ecosystem properties. And I think it's misleading to say in the abstract that "phylogenetic alpha diversity declines for all taxonomic groups", even with the particular metric in parentheses. The truth is that total PD did just the opposite. The fact that total PD and richness are correlated has no bearing on the question of whether total PD or MPD are more or less relevant when interested either in ecosystem function or the conversation of biodiversity more generally. Reporting both sets of results in the main figures and discussing them in the text with appropriate precision and nuance would be easy to do.

(2b) Line 395: Related to the last point, I don't see how these results demonstrate the importance of using richness-independent measures of PD. Just because something different was found doesn't make these any more appropriate for a particular objective. They demonstrate that different metrics behave in different ways, not that it's more important to use one that diverges from taxonomic results.

(3) Effect sizes. The authors stress the importance of results being significantly different from zero, and much less so the effect sizes. But some effect sizes seem really tiny, especially for phylogenetic results. People have asked how phylogenetic diversity influences things like ecosystem function. Do changes of <1% matter in any way that's been measured? Is there any plausible scenario in which a change of <1% might matter for conservation or anything else? For other metrics, changes are on the order of 5-10%, which is larger, but again - do changes in richness of this magnitude matter in studies that have looked at the *effects* of richness? I think the answer is 'no'. So while these changes are interesting, and maybe larger changes are coming, the results as they are seem unlikely to be indicative of other consequences in communities or ecosystems. The manuscript suggests otherwise.

(4) Line 149: I'm not sure what's mean by the "sample effort corrected measurement" here.

(5) Line 157: "management" is pretty vague, and could presumably include something like controlled grazing.

(6) Line 202: Were species removed also from the taxonomic analyses? The justification for not using them in phylogenetic analyses is obvious, but there's no reason to eliminate them from analyses where all necessary information is present.

(7) Line 245: Do you mean `dat_2`? If it's a dissimilarity, then it would be a lower value in `dat_2` that indicates homogenization. Although the description makes it sound like a similarity. Something here should be clarified.

(8) Line 262: Log ratios are commonly used for variables with no upper bound, but it seems unusual for indices of beta diversity. Effectively it's a proportional change in a number that itself is bounded between zero and one. I don't have anything specific to suggest here apart from thinking about whether a log ratio or a difference makes more ecological sense. A statistical expert might have advice.

(9) Line 362: I don't see any way in which these new results "provide an explanation" for earlier results. They simply reinforce what was already found, without digging deeper as to why. That said, the most parsimonious explanation is that one type of data focuses 100% on land-use impacts, while the other type of data focuses on temporal changes that could have occurred for any one of many reasons. And that we knew already anyway.

Referee: 2

Comments to the Author(s)

This paper aims to assess how biodiversity is changing at various spatial scales around the globe. The authors use existing data on birds, fish and plants to assess changes in α and β diversity. They used measures of species richness as well as phylogenetic measures of diversity (independent of richness) to investigate how multiple aspects of diversity are changing. The datasets that were used in the paper could be of 3 forms: 1) the data could include land-use conversion studies where species richness varied as a function of land-use change, 2) the data could include contemporary species list data where knowledge is available on which species are non-native (i.e. have arrived recently), and others which have gone extinct, and 3) time-series data.

I thought the paper was well-written and the data analysis and figures were appropriate. I have two main, fairly major concerns, and other smaller critiques. I hope that the authors can address the major concerns with either more discussion or further analysis of data.

My first major concern is that there have been a number of studies recently investigating how patterns of diversity are changing (or not), and yet none of them (including this paper) overcome the most limiting factor of understanding how biodiversity is changing across the globe: namely huge spatial sampling bias, and general lack of representation of changes in biodiversity outside of North America, central Europe and a few other exceptional locations (e.g. South Africa, Argentina, and the West Indies). This is well-represented in their Figure 1. Much of the globe is not even represented by a single study, and a large fraction of the terrestrial land surface is represented by a 1-2 datasets. The poor sample size across the globe and hugely biased spatial sampling limits any general statement about what is happening to biodiversity. I think that much more emphasis should be placed on this result: we know very little (in some cases literally nothing) about what is happening to biodiversity. I think the authors should highlight the fact that the general data quantity and distribution is poor, with a greater emphasis placed on the fact that they cannot get estimates, nor confidence intervals around estimates of biodiversity change in many regions of the globe. I would also like to know what the authors recommend to alleviate the situation. How could biodiversity estimates could be generated in regions where sampling programs are not under way and likely won't be initiated in the near future?

Secondly, the three different ways of analyzing biodiversity data appear to have different assumptions and limitations and it would be nice if there were a bit more discussion or perhaps analysis of how this might affect interpretation. The time-series analysis is the least problematic, because presumably estimates of biodiversity are generated in the same way over time. However, I'd be curious to know whether this type of monitoring is done more frequently in pristine or restored sites, rather than in sites undergoing major land-use transitions. If so, then this may bias estimates from this type of data to show positive trends in diversity (as seen in Figure 3). Alternatively, how are extinctions detected in the Species List data? If extinctions require a higher

burden of proof (lack of observation for a certain time period), than invasions, this may also bias analysis towards estimates of biodiversity increase. Lastly, the land-use conversions included in the paper only include conversions from a relatively natural to a human-impacted state. This may bias estimates towards findings of biodiversity losses. However, this type of analysis should be standardized or weighted by the prevalence of particular land-use transitions observed within a biome, bioclimatic region, or across the globe. This type of analysis would then not bias our view of biodiversity change simply due to a focus on particular land-use transitions of interest.

Specific minor concerns.

P. 8, L. 133: Doesn't removing countries with <10 non-native records across basins bias your analysis towards observations of increasing diversity?

P. 9, L. 170: Can you familiarize us with the assumptions behind this method of inserting extinct birds into the stem preceding the most recent common ancestor?

P. 10, L. 192: How can this decision to equally space nodes be justified? What impact might it have?

P. 11, L. 202: Did this more strongly affect dat_1 or dat_2? Or were removals randomly or equally distributed between both datasets?

General question about the results: Can you confirm that all sites/studies were investigating locations where human influence was present. i.e. not pristine or restored sites? How representative are the land-use conversion types as a percentage of the types of conversion that we see across the globe?

It's disconcerting that the different data types produce different insights. It suggests that they represent different regions or are differentially reflective of different types of biodiversity change.

P. 16, L. 327: Please spell out the difference between extant and established species.

P. 17, L. 334: Assumes that non-native species were not there 20 years or more ago. Some may be non-native but already well-naturalized.

P. 18, L. 366: Why wouldn't such information be reliable?

P. 18, L. 370: This is interesting, and likely robust.

P. 19, L. 376: The link between phylogenetic and functional diversity is not universal and therefore shouldn't be used without support.

P. 20, L. 400: Why didn't you do this? Or at least present the data on how spatially biased the different types of data are? Land-use: how well do these land-use conversions reflect what's happening globally? Species lists: Are species generally gained only because extinctions are hard to document? Time-series: Are these spatially biased? Occurring in pristine or restored, or largely protected environments?

Author's Response to Decision Letter for (RSPB-2019-2404.R0)

See Appendix A.

RSPB-2020-0777.R0

Review form: Reviewer 1

Recommendation

Major revision is needed (please make suggestions in comments)

Scientific importance: Is the manuscript an original and important contribution to its field?
Good

General interest: Is the paper of sufficient general interest?
Excellent

Quality of the paper: Is the overall quality of the paper suitable?
Good

Is the length of the paper justified?
Yes

Should the paper be seen by a specialist statistical reviewer?
No

Do you have any concerns about statistical analyses in this paper? If so, please specify them explicitly in your report.
No

It is a condition of publication that authors make their supporting data, code and materials available - either as supplementary material or hosted in an external repository. Please rate, if applicable, the supporting data on the following criteria.

Is it accessible?
No

Is it clear?
Yes

Is it adequate?
No

Do you have any ethical concerns with this paper?
No

Comments to the Author

Overall, the authors have responded comprehensively to all of my earlier criticisms, and apart from little quibbles one might think of, I think the justification in most cases is strong. There is, however, one exception where I think the rationale is faulty: how to analyze land-use effects on beta diversity.

How to assess the effects of land use on beta diversity? The authors rightly point out that comparing beta diversity for dat_1 (unmodified habitat) with dat_1 + dat_2 (unmodified + modified) requires an assumption that dat_1 is a reasonable representation of the original landscape. They use this as justification for focusing on the comparison of dat_1 and dat_2. However, comparing dat_1 to dat_2 requires precisely the same assumption (dat_1 = "historical", dat_2 = "current") PLUS an additional and much less likely assumption that an entire landscape has been converted from one habitat to another. The latter is representative of few large areas on earth, and by definition it is representative of literally none of the landscapes used in the paper (since they have data for both modified and unmodified habitats). In short, to answer the authors implicit query, I disagree quite strongly with the rationale. The choice of which to present is rather important, since the two options lead to completely different conclusions. As such, it seems important to at least present in the main paper (not buried in an appendix, and certainly not relegated only to a cover letter) analyses of both scenarios, and if one were to be chosen it should be the one with fewer unrealistic assumptions, not more. As it is, the

paper communicates only the less likely conclusion, without even a hint at the more likely conclusion.

Decision letter (RSPB-2020-0777.R0)

09-May-2020

Dear Dr Li,

Thank you for submitting a revised version of your manuscript. This has now been reviewed by one of the referees on the original version, and the reviewer's comments are included at the end of this email.

As you will see, the reviewer is generally happy with the revised version of the manuscript. The one exception is with regard to the comparison of unmodified and modified habitats, where they point out the limitations of the approach taken. The reviewer also notes that the data are not fully accessible. We would therefore like to invite you to revise your manuscript to address these issues.

Research ethics:

Use of animals and field studies:

If your study uses animals please include details in the methods section of any approval and licences given to carry out the study and include full details of how animal welfare standards were ensured. Field studies should be conducted in accordance with local legislation; please

include details of the appropriate permission and licences that you obtained to carry out the field work.

Please submit a copy of your revised paper within three weeks. If you are unable to meet this deadline, especially given the current circumstances, please get in touch as soon as possible to discuss an extension.

Finally, I would also like to thank you for such a thorough, well-written and courteous 'response to reviewers'; it was a pleasure to read.

I hope you and your co-authors are well in these challenging times.

Best wishes,
Professor Loeske Kruuk
<mailto:proceedingsb@royalsociety.org>

Reviewer(s)' Comments to Author:

Referee: 1

Comments to the Author(s).

Overall, the authors have responded comprehensively to all of my earlier criticisms, and apart from little quibbles one might think of, I think the justification in most cases is strong. There is, however, one exception where I think the rationale is faulty: how to analyze land-use effects on beta diversity.

How to assess the effects of land use on beta diversity? The authors rightly point out that comparing beta diversity for dat_1 (unmodified habitat) with dat_1 + dat_2 (unmodified + modified) requires an assumption that dat_1 is a reasonable representation of the original landscape. They use this as justification for focusing on the comparison of dat_1 and dat_2. However, comparing dat_1 to dat_2 requires precisely the same assumption (dat_1 = "historical", dat_2 = "current") PLUS an additional and much less likely assumption that an entire landscape has been converted from one habitat to another. The latter is representative of few large areas on earth, and by definition it is representative of literally none of the landscapes used in the paper (since they have data for both modified and unmodified habitats). In short, to answer the authors implicit query, I disagree quite strongly with the rationale. The choice of which to present is rather important, since the two options lead to completely different conclusions. As such, it seems important to at least present in the main paper (not buried in an appendix, and certainly not relegated only to a cover letter) analyses of both scenarios, and if one were to be chosen it should be the one with fewer unrealistic assumptions, not more. As it is, the paper communicates only the less likely conclusion, without even a hint at the more likely conclusion.

Author's Response to Decision Letter for (RSPB-2020-0777.R0)

See Appendix B.

RSPB-2020-0777.R1 (Revision)

Review form: Reviewer 1

Recommendation

Accept as is

Scientific importance: Is the manuscript an original and important contribution to its field?

Good

General interest: Is the paper of sufficient general interest?

Excellent

Quality of the paper: Is the overall quality of the paper suitable?

Excellent

Is the length of the paper justified?

Yes

Should the paper be seen by a specialist statistical reviewer?

No

Do you have any concerns about statistical analyses in this paper? If so, please specify them explicitly in your report.

No

It is a condition of publication that authors make their supporting data, code and materials available - either as supplementary material or hosted in an external repository. Please rate, if applicable, the supporting data on the following criteria.

Is it accessible?

Yes

Is it clear?

Yes

Is it adequate?

Yes

Do you have any ethical concerns with this paper?

No

Comments to the Author

I appreciate the authors' willingness to include both sets of results. We can agree to disagree on whether the landscape approach represents an "exception" with an "assumption (that) does not likely hold in most landscapes", or if it represents a more realistic assessment based on an assumption more likely to hold than the alternative.

Decision letter (RSPB-2020-0777.R1)

21-May-2020

Dear Dr Li

I am pleased to inform you that your manuscript entitled "Changes in Taxonomic and Phylogenetic Diversity in the Anthropocene" has been accepted for publication in Proceedings B.

If you are likely to be away from e-mail contact, or if you have any problems checking the proofs given the current circumstances, please let us know. Due to rapid publication and an extremely tight schedule, if comments are not received, we may publish the paper as it stands.

Open Access

Paper charges

Thank you for your excellent contribution. On behalf of the Editors of the Proceedings B, we look forward to your continued contributions to the Journal.

Yours sincerely,

Professor Loeske Kruuk

Associate Editor:

Board Member: 1

Comments to Author:

(There are no comments.)

Board Member: 2

Comments to Author:

(There are no comments.)

Appendix A

Response to reviewers

Daijiang Li, Julian D. Olden, Julie L. Lockwood, Sydne Record,
Michael L. McKinney, and Benjamin Baiser

08 April, 2020

13-Dec-2019

Dear Dr Li,

I am writing to inform you that your manuscript RSPB-2019-2404 entitled “Changes in Taxonomic and Phylogenetic Diversity in the Anthropocene” has, in its current form, been rejected for publication in Proceedings B.

This action has been taken on the advice of referees, who have recommended that substantial revisions are necessary. We are all agreed that this is a very interesting topic and an impressive data set, but the reviewers have identified several major issues. With this in mind we would be happy to consider a resubmission, provided the comments of the referees are fully addressed. However please note that this is not a provisional acceptance.

- 1) A ‘response to referees’ document including details of how you have responded to the comments, and the adjustments you have made.
- 2) A clean copy of the manuscript and one with ‘tracked changes’ indicating your ‘response to referees’ comments document.
- 3) Line numbers in your main document.

To upload a resubmitted manuscript, log into <http://mc.manuscriptcentral.com/prsb> and enter your Author Centre, where you will find your manuscript title listed under “Manuscripts with Decisions.” Under “Actions,” click on “Create a Resubmission.” Please be sure to indicate in your cover letter that it is a resubmission, and supply the previous reference number.

Please also check the accessibility and adequacy of your archived data files, as one of the reviewers commented that these were not sufficient.

Yours sincerely,

Professor Loeske Kruuk
Editor

Associate Editor

Board Member: 1

Comments to Author:

The authors have put together an impressive dataset and applied some interesting analyses, including phylogenetic ones that make this a paper with lots of potential. The reviewers have suggested major revisions and made some important points which may require some re-analysis. In particular the authors should reconsider how their land-use conversions are incorporated into the analyses as well as their conclusions from the phylogenetic diversity metrics. Other important changes have been recommended as well and I look forward to seeing a revised version of the manuscript.

Dear Editors,

Thank you very much for your assistance with our manuscript. We are very pleased that the reviewers found value in our study and described it as making “an important contribution to the literature” and that “the paper was well-written”. The reviewers’ comments are extremely helpful to improve the quality and clarity of our manuscript.

The two major critiques of our manuscript were the following. First, a reviewer questioned the use of phylogenetic alpha and beta diversity metrics that were independent of species richness in our main text. Second, a reviewer requested a more in-depth discussion of the caveats in our study largely stemming from the heterogeneous nature of our data. We have now conducted a comprehensive revision of the manuscript to address these two overarching concerns, as well as all the minor comments raised by the reviewers.

We address the first critique by moving the results of common phylogenetic alpha (Faith’s PD) and beta (PhyloSor) diversity metrics that are dependent on species richness into the main figures and Results of the manuscript. We further discuss the implications of these different metrics in the Discussion. We address the second critique by carefully reworking the Discussion to clearly detail specific biases of each data type and describe possible remedies for the dearth of data in different geographic locations. The take-home message of this new investigation is now stated in the Discussion; specifically, our main result “... holds across different taxonomic groups and different data types, suggesting a strongly coherent global pattern.”

After adding new Results and Discussion materials, the manuscript length is now over the limit allowed by Proc. B. We therefore have moved the details of phylogenies building into the Supplementary.

We feel that the significant revisions made to our manuscript have resulted in a much stronger study and contribution to the literature. It was only possible due to the constructive comments from both the Editor and reviewers.

Thank you for considering our manuscript.

Daijiang (on behalf of all authors)

Reviewer(s)' Comments to Author:

Referee: 1

Comments to the Author(s)

This is an interesting and well-written manuscript putting together several threads from the literature concerning biodiversity change: how results vary among data sources, how alpha and beta diversity change simultaneously, and whether phylogenetic diversity changes mirror those found for taxonomic diversity. The authors use an impressive combination of data sets for multiple taxa and geographic regions to tackle these questions. The analyses include quite a few steps involving data subsetting, choices of metrics, and model structure; most justifications seems fine, although there are some areas where I think important conclusions depend on questionable decisions (described below). The results to a considerable extent confirm what is already known: land use can cause alpha diversity declines (as revealed by space-for-time datasets); other kinds of changes influencing biodiversity do not cause declines on average (revealed in other data sets); and biotic homogenization is common. The phylogenetic results are definitely a new addition, and having all of these things in one place is useful as well. Overall, I think this study is likely to make an important contribution to the literature on this topic.

Note: I am not knowledgeable enough of phylogenetic methods to comment on that; I also can't keep track of all the dos and don'ts of statistical practice, although the use of random effects for categories that sure seem like fixed effects, or with very small numbers of categories, seems to violate common advice on best practices.

The reviewer has made an important point that we welcome the opportunity to address. Whether to include a categorical variable as a fixed term or random term depends largely on the purpose of the study. Here, our focal interest is on patterns of biodiversity change instead of comparing differences between the levels of categorical variables. Therefore, including these categorical variables as random terms make more sense from a philosophical point of view. Technically, including a categorical variable as a random term performs equally to a classical regression with such a variable as a fixed term (page 275-276 Gelman and Hill 2006).

- (1) Land use and biotic homogenization. For the Resample and Species List data sets, I think the inferences about biotic homogenization are valid, in that particular set of places become more similar over time. For the Land Use data, I think there is a flaw in the logic. When one converts natural habitats to urban areas or crops (for example), it is not the case that a landscape is now entirely urban or agricultural. Rather, the modified landscape is a mix of these different land cover types. So, asking how land use has altered beta diversity requires that `dat_2` (the data matrix representing "after") include not only those plots under a given land use, but all plots. If the species composition of crop fields is different from that in natural vegetation, then one may well expect that beta diversity will have done up at the landscape scale, although the outcome will depend on variation both within and among land cover types. In short, if you think about the places where these land use studies are done, people can create land cover types for which replicate plots are compositionally very similar to one another, while at the same time creating a landscape that is not more homogenous but just the opposite. Looking at

things both ways would be quite straightforward with the data at hand, and provide a richer picture of how land use has influenced beta diversity.

We agree with the reviewer that including both `dat_1` (natural) and `dat_2` (changed) plots to calculate beta diversity after land use change is required. However, to compare with beta diversity before land use change, it requires that we also have information about the baseline conditions of plots in `dat_2` (i.e. we need both `dat_1` and `base_dat_2` to make the comparison fair).

Take the hypothetical example of a 10-by-10 km natural area that is divided into 100 1 km² cells. Then, suppose that the land uses of half of these cells were changed to farmland, for instance. In this case, 50 cells that have not been changed form `dat_1` and the other 50 cells form `dat_2`. If we calculate beta diversity with all 100 cells as the beta diversity after land use change, then we necessarily require the information of all 100 cells before land use change to recover the original beta diversity of this region. Unfortunately, such information is rarely available (i.e. we rarely know information of `dat_2` before it is converted to other land type). If only part of these 100 cells were converted, then the overall beta diversity will depend on the proportion of converted area (see a hypothetical figure below). An alternative approach is to just compare `dat_1` and `dat_2` to estimate how the spatial beta diversity would change if the entire area was converted to another land type.

With the data available to us, it is possible to compare `dat_1` with `dat_1 + dat_2` of land use data (results in the table below). Doing this, we see that the taxonomic beta diversity increased as expected. However, given that we lack baseline information for sites that have been converted to other land types, we focused on comparing `dat_1` and `dat_2` in the main text (i.e. space-for-time substitution). We would be happy to entertain the prospects of adding these results to the Appendix if the reviewer and editor disagree with our rationale.

	Estimate LRR	Change %	lower LRR	upper LRR
Sor_turnover	0.076	+6.89	0.036	0.117
pSor_turnover	0.051	+5.23	0.017	0.084
PCDp	0.005	-0.50	-0.006	0.017

(2a) Metrics of phylogenetic diversity. I think it's great that the authors recognize and report analyses of phylogenetic diversity both "overall" (i.e., metrics correlated with richness), and a component that is independent of richness. However, the decision to focus the main analyses on richness-independent measures has important consequences for the validity of arguments linking PD to things like resilience, stability, and ecosystem function (lines 54, 377), and even for the generality of any conclusion concerning PD in general. For example, a typical situation in many Species List data sets seems to be few extinctions and a bunch of additions; for sake of argument, imagine that there are only additions. With the addition of new species, the total amount of the tree of life represented can only go up. But if those new species are closely related to residents, something like MPD can go down. I know of no evidence indicating that a change in MPD via this pathway will compromise the stability or functioning of an ecosystem. The kinds of metrics that are correlated with taxonomic richness are at least as likely, and perhaps even more likely, to predict altered ecosystem properties. And I think it's misleading to say in the abstract that "phylogenetic alpha diversity declines for all taxonomic groups", even with the particular metric in parentheses. The truth is that total PD did just the opposite. The fact that total PD and richness are correlated has no bearing on the question of whether total PD or MPD are more or less relevant when interested either in ecosystem function or the conversation of biodiversity more generally. Reporting both sets of results in the main figures and discussing them in the text with appropriate precision and nuance would be easy to do.

We appreciate this suggestion and have now reported both sets of results in the main text in the revision.

(2b) Line 395: Related to the last point, I don't see how these results demonstrate the importance of using richness-independent measures of PD. Just because something different was found doesn't make these any more appropriate for a particular objective. They demonstrate that different metrics behave in different ways, not that it's more important to use one that diverges from taxonomic results.

We have removed this part of the text given that we now report both sets of results in the main text.

(3) Effect sizes. The authors stress the importance of results being significantly different from zero, and much less so the effect sizes. But some effect sizes seem really tiny, especially for phylogenetic results. People have asked how phylogenetic diversity influences things like ecosystem function. Do changes of <1% matter in any way that's been measured? Is there any plausible scenario in which a change of <1% might matter for conservation or anything else? For other metrics, changes are on the order of 5-10%, which is larger, but again – do changes in richness of this magnitude matter in studies that have looked at the effects of richness? I think the answer is 'no'. So while these changes are interesting, and maybe larger changes

are coming, the results as they are seem unlikely to be indicative of other consequences in communities or ecosystems. The manuscript suggests otherwise.

This is a great question regarding whether a particular magnitude of changes in biodiversity affects ecosystem functions. Indeed, this is an actively growing topic of interest in science, and unfortunately addressing it comprehensively is beyond the scope of our study. However, the general consensus in the literature suggests that biodiversity (both alpha and beta) has a positive relationship with ecosystem functions. In the revision, we avoid linking changes in biodiversity with consequences in ecosystem functions in the Results; though we briefly mentioned it in the Introduction and the Discussion.

(4) Line 149: I'm not sure what's mean by the "sample effort corrected measurement" here.

The PREDICTS dataset contains a column of sample effort corrected by measurement (calculated by dividing the diversity observations by sampling effort [e.g., number of traps, number of sampling events, length of sampling transect]). Please see page 161 of Hudson et al. (2017) for details.

(5) Line 157: "management" is pretty vague, and could presumably include something like controlled grazing.

Yes, it could. Here we use "management" in the broadest sense to refer to any action by humans to attempt to change the environment to achieve a particular desired outcome.

(6) Line 202: Were species removed also from the taxonomic analyses? The justification for not using them in phylogenetic analyses is obvious, but there's no reason to eliminate them from analyses where all necessary information is present.

Yes, species were removed from all analyses if they were not contained in the phylogeny. The main reason to do so is to make sure both taxonomic diversity and phylogenetic diversity results are from the same community data and thus are comparable. Rerunning taxonomic analyses with all species did not change our main results and conclusions.

(7) Line 245: Do you mean `dat_2`? If it's a dissimilarity, then it would be a lower value in `dat_2` that indicates homogenization. Although the description makes it sound like a similarity. Something here should be clarified.

Yes, we mean `dat_2`. Thanks for pointing this out.

(8) Line 262: Log ratios are commonly used for variables with no upper bound, but it seems unusual for indices of beta diversity. Effectively it's a proportional change in a number that itself is bounded between zero and one. I don't have anything specific to suggest here apart from thinking about whether a log ratio or a difference makes more ecological sense. A statistical expert might have advice.

You are correct that log ratios are commonly used for variables that are larger than 0 and with no upper bound. It is also true that beta diversity normally is bounded between 0 and 1. However, the ratio of beta diversity can be any positive value with no upper bound. Therefore, we think the log ratio of beta diversity is valid here.

- (9) Line 362: I don't see any way in which these new results "provide an explanation" for earlier results. They simply reinforce what was already found, without digging deeper as to why. That said, the most parsimonious explanation is that one type of data focuses 100% on land-use impacts, while the other type of data focuses on temporal changes that could have occurred for any one of many reasons. And that we knew already anyway.

We have revised this sentence to "Our results thus confirm the current divergence in results of contemporary changes in taxonomic species richness".

Referee: 2

Comments to the Author(s)

This paper aims to assess how biodiversity is changing at various spatial scales around the globe. The authors use existing data on birds, fish and plants to assess changes in α and β diversity. They used measures of species richness as well as phylogenetic measures of diversity (independent of richness) to investigate how multiple aspects of diversity are changing. The datasets that were used in the paper could be of 3 forms: 1) the data could include land-use conversion studies where species richness varied as a function of land-use change, 2) the data could include contemporary species list data where knowledge is available on which species are non-native (i.e. have arrived recently), and others which have gone extinct, and 3) time-series data.

I thought the paper was well-written and the data analysis and figures were appropriate. I have two main, fairly major concerns, and other smaller critiques. I hope that the authors can address the major concerns with either more discussion or further analysis of data.

Thank you for your helpful comments.

My first major concern is that there have been a number of studies recently investigating how patterns of diversity are changing (or not), and yet none of them (including this paper) overcome the most limiting factor of understanding how biodiversity is changing across the globe: namely huge spatial sampling bias, and general lack of representation of changes in biodiversity outside of North America, central Europe and a few other exceptional locations (e.g. South Africa, Argentina, and the West Indies). This is well-represented in their Figure 1. Much of the globe is not even represented by a single study, and a large fraction of the terrestrial land surface is represented by a 1-2 datasets. The poor sample size across the globe and hugely biased spatial sampling limits any general statement about what is happening to biodiversity. I think that much more emphasis should be placed on this result: we know very little (in some cases literally nothing) about what is happening to biodiversity. I think the authors should highlight the fact that the general data quantity and distribution is poor, with a greater emphasis placed on the fact that they cannot get estimates, nor confidence intervals around estimates of biodiversity change in many regions of the globe. I would also like to know what the authors recommend to alleviate the situation. How could biodiversity estimates could be generated in regions where sampling programs are not under way and likely won't be initiated in the near future?

We appreciate this thoughtful comment. The issue of geographical disparity in biodiversity data is a common challenge of all current macroecological studies. Here, we focus on the question that is tractable based on the extensive datasets that we have collated over the past several years, with the aim to synthesize the overall patterns of changes in taxonomic and phylogenetic biodiversity. We have revised the manuscript, especially the Discussion section, extensively to clearly acknowledge all limitations and articulate what we believe are constructive ways forward.

Secondly, the three different ways of analyzing biodiversity data appear to have different assumptions and limitations and it would be nice if there were a bit more discussion or perhaps analysis of how this might affect interpretation. The time-series analysis is the least problematic, because presumably estimates of biodiversity are generated in the same way over time. However, I'd be curious to know whether this type of monitoring is done more frequently in pristine or restored sites, rather than in sites undergoing major land-use transitions. If so, then this may bias estimates from this type of data to show positive trends in diversity (as seen in Figure 3). Alternatively, how are extinctions detected in the Species List data? If extinctions require a higher burden of proof (lack of observation for a certain time period), than invasions, this may also bias analysis towards estimates of biodiversity increase. Lastly, the land-use conversions included in the paper only include conversions from a relatively natural to a human-impacted state. This may bias estimates towards findings of biodiversity losses. However, this type of analysis should be standardized or weighted by the prevalence of particular land-use transitions observed within a biome, bioclimatic region, or across the globe. This type of analysis would then not bias our view of biodiversity change simply due to a focus on particular land-use transitions of interest.

These are all excellent points. We have discussed these limitations in the extensively revised Discussion. Briefly, for the survey-resurvey datasets, almost all of them were conducted in pristine sites and none of them were in sites undergoing major land-use transitions. For the Species List Data, detecting extinctions is definitely a big challenge. We have completed the Species List with potential extinction records from the literature as best as we can (see Methods). For Land Use Data, we fully agree that different land types should be weighted by their prevalence within a region whenever possible. However, getting data about prevalence of different land use changes currently is still a big challenge itself and we lack the capacity to do it here.

Specific minor concerns.

P. 8, L. 133: Doesn't removing countries with <10 non-native records across basins bias your analysis towards observations of increasing diversity?

Yes, it does. However, if we include countries with very small numbers of non-native records it would dilute the effects of species invasions on biodiversity and likely will bias analysis towards producing small changes in biodiversity. Therefore, we think that our decision to remove countries with <10 records is a reasonable cutoff.

P. 9, L. 170: Can you familiarize us with the assumptions behind this method of inserting extinct birds into the stem preceding the most recent common ancestor?

Here, the assumption is that the extinct birds and the clade selected are closely related. Inserting the extinct birds into the stem preceding their most recent common ancestor will give the least conservative estimate of phylogenetic diversity for this clade (i.e. maximum total branch length).

P. 10, L. 192: How can this decision to equally space nodes be justified? What impact might it have?

According to the authors of the `bladj` program that we used, without more information, estimating node ages by equally placing ages between nodes with dates is a reasonable choice. For the purpose of calculating phylogenetic diversity, it actually provides very close results with those derived from phylogenies based on gene sequence data (Li et al. 2019).

P. 11, L. 202: Did this more strongly affect `dat_1` or `dat_2`? Or were removals randomly or equally distributed between both datasets?

This is an excellent question. We tallied the removals and found the median percentage of removed species in `dat_1` and `dat_2` is 3.6% and 3.7%, respectively, indicating these did not bias our results.

General question about the results: Can you confirm that all sites/studies were investigating locations where human influence was present. i.e. not pristine or restored sites? How representative are the land-use conversion types as a percentage of the types of conversion that we see across the globe? It's disconcerting that the different data types produce different insights. It suggests that they represent different regions or are differentially reflective of different types of biodiversity change.

This is a helpful insight. However, gathering this data worldwide is beyond the scope of our manuscript. We did include this concern in the Discussion as a caveat to our results.

P. 16, L. 327: Please spell out the difference between extant and established species.

Here, all established species are non-native species while all extant species are native species. We have revised the manuscript to make this clear.

P. 17, L. 334: Assumes that non-native species were not there 20 years or more ago. Some may be non-native but already well-naturalized.

Correct.

P. 18, L. 366: Why wouldn't such information be reliable?

There are several reasons for this. For example, it is possible that the data available are too old and thus will not reflect the current situations; or the data are generated based on too coarse of a spatial resolution. Anyway, we have removed this word in the revision.

P. 18, L. 370: This is interesting, and likely robust.

We agree.

P. 19, L. 376: The link between phylogenetic and functional diversity is not universal and therefore shouldn't be used without support.

It is true that the link between phylogenetic and functional diversity is not universal though recent simulations suggest a positive relationship between them (Tucker et al. 2018). Here, we think it is appropriate to write "suggesting potential declines ...".

P. 20, L. 400: Why didn't you do this? Or at least present the data on how spatially biased the different types of data are? Land-use: how well do these land-use conversions reflect what's happening globally? Species lists: Are species generally gained only because extinctions are hard to document? Time-series: Are these spatially biased? Occurring in pristine or restored, or largely protected environments?

We did not weight different land types with their corresponding land areas because we don't have such data. For the other questions, please see the above responses and the revised manuscript.

References

- Hudson, L. N., T. Newbold, S. Contu, S. L. Hill, I. Lysenko, A. De Palma, H. R. Phillips, T. I. Alhusseini, F. E. Bedford, D. J. Bennett, and others. 2017. The database of the predicts (projecting responses of ecological diversity in changing terrestrial systems) project. *Ecology and Evolution* 7:145–188.
- Li, D., L. Trotta, H. E. Marx, J. M. Allen, M. Sun, D. E. Soltis, P. S. Soltis, R. P. Guralnick, and B. Baiser. 2019. For common community phylogenetic analyses, go ahead and use synthesis phylogenies. *Ecology* 100:e02788.
- Tucker, C. M., T. J. Davies, M. W. Cadotte, and W. D. Pearse. 2018. On the relationship between phylogenetic diversity and trait diversity. *Ecology* 99:1473–1479.

Appendix B

09-May-2020

Dear Dr Li,

Thank you for submitting a revised version of your manuscript. This has now been reviewed by one of the referees on the original version, and the reviewer's comments are included at the end of this email.

As you will see, the reviewer is generally happy with the revised version of the manuscript. The one exception is with regard to the comparison of unmodified and modified habitats, where they point out the limitations of the approach taken. The reviewer also notes that the data are not fully accessible. We would therefore like to invite you to revise your manuscript to address these issues.

Research ethics:

Use of animals and field studies:

If you wish to submit your data to Dryad (<http://datadryad.org/>) and have not already done so you can submit your data via this link [http://datadryad.org/submit?journalID=RSPB&manu=\(Document not available\)](http://datadryad.org/submit?journalID=RSPB&manu=(Document%20not%20available)), which will take you to your unique entry in the Dryad repository.

Please submit a copy of your revised paper within three weeks. If you are unable to meet this deadline, especially given the current circumstances, please get in touch as soon as possible to discuss an extension.

Finally, I would also like to thank you for such a thorough, well-written and courteous 'response to reviewers'; it was a pleasure to read.

I hope you and your co-authors are well in these challenging times.

Best wishes,

Professor Loeske Kruuk
mailto:proceedingsb@royalsociety.org

Dear Editors,

Thank you for allowing us to revise our manuscript. In this revision, we address all concerns by 1) including diversity results based on both modified and unmodified datasets and 2) depositing the data supporting our results into Dryad (<https://datadryad.org/stash/share/zPLkMD4xMZwFc9xEWwAGun1y05PtDgTb9g8ip4gOHNg>). We have included the DOI of the data repository in the Data availability section; it is also cited as a reference.

All the best,
Daijiang (on behalf of all authors)

Reviewer(s)' Comments to Author:

Referee: 1

Comments to the Author(s).

Overall, the authors have responded comprehensively to all of my earlier criticisms, and apart from little quibbles one might think of, I think the justification in most cases is strong. There is, however, one exception where I think the rationale is faulty: how to analyze land-use effects on beta diversity.

How to assess the effects of land use on beta diversity? The authors rightly point out that comparing beta diversity for dat_1 (unmodified habitat) with dat_1 + dat_2 (unmodified + modified) requires an assumption that dat_1 is a reasonable representation of the original landscape. They use this as justification for focusing on the comparison of dat_1 and dat_2. However, comparing dat_1 to dat_2 requires precisely the same assumption (dat_1 = "historical", dat_2 = "current") PLUS an additional and much less likely assumption that an entire landscape has been converted from one habitat to another. The latter is representative of few large areas on earth, and by definition it is representative of literally none of the landscapes used in the paper (since they have data for both modified and unmodified habitats). In short, to answer the authors implicit query, I disagree quite strongly with the rationale. The choice of which to present is rather important, since the two options lead to completely different conclusions. As such, it seems important to at least present in the main paper (not buried in an appendix, and certainly not relegated only to a cover letter) analyses of both scenarios, and if one were to be chosen it should be the one with fewer unrealistic assumptions, not more. As it is, the paper communicates only the less likely conclusion, without even a hint at the more likely conclusion.

Thanks for your help with our manuscript. We have now reported the diversity results based on dat_1 + dat_2 in the revised manuscript. We have also updated relevant parts in the Methods and Discussion sections.